# Is Long COVID a State of Systemic Pericyte Disarray?

**DOI:** 10.3390/jcm11030572

**Published:** 2022-01-24

**Authors:** Olcay Y. Jones, Sencer Yeralan

**Affiliations:** 1Pediatric Rheumatology, Walter Reed National Military Medical Center, Bethesda, MD 20889, USA; 2School of IT and Engineering, ADA University, Baku AZ1008, Azerbaijan

**Keywords:** Long COVID, pericyte, Triple-A, complex dynamical systems, entropy, emergent systems intelligence

## Abstract

The most challenging aspect of Post-Acute Sequelae of SARS-CoV-2 Infection (PASC) or Long COVID remains for the discordance between the viral damage from acute infection in the recent past and susceptibility of Long COVID without clear evidence of post infectious inflammation or autoimmune reactions. In this communication we propose that disarray of pericytes plays a central role in emerge of Long COVID. We assume pericytes are agents with “Triple-A” qualities, i.e., analyze-adapt and advance, necessary for sustainability of host homeostasis. Based on this view, we further suggest Long COVID may provide a model system to integrate system theory and complex adaptive systems to explore a new class of maladies those are currently not well defined and with no remedies.

## 1. Defining Long COVID

Long COVID, a.k.a. Post COVID-19, and Post-Acute Sequelae of SARS-CoV-2 Infection (PASC) is a clinical case definition based on a plethora of signs and symptoms lasting for more than 4 weeks following COVID-19 [1,2,3]. It is a diagnosis of exclusion, without a specific test or known biomarker and has two requirements: evidence of SARS-CoV-2 exposure by history or serology, and the absence of ongoing SARS-CoV-2 viral load or any other infectious agents or etiologies that can account for impaired health.

Long COVID is a post viral state of health affecting about 20% of COVID-19 patients. It manifests almost always with extreme fatigue and low endurance [4]. Most suffer from symptoms of nervous system dysfunction. These include autonomic dysfunction with palpitation, tachycardia, postural hypotension, and GI-dysmotility; Cognitive concerns with brain fog, headaches, difficulty in concentration; emotional liability with depression, anxiety, and poor sleep [5]. In addition, patients often describe shortness of breath and dyspnea, arthralgia, myofascial pain and weakness. These are chronic complains that are entangled with low stamina with day-to-day fluctuations of small gains followed by severe exhaustion requiring prolonged recovery times. In general, all affected complain of their poor quality life due to mild (self-sufficient) to severe (bed-ridden) functional impairment, affecting productivity and well-being. Overall, about 50% of all Long COVID patients have been unable to reach their baseline health even after a year post-infection. Currently there is no effective treatment of Long COVID, as the understanding of its pathogenesis remains elusive [6].

## 2. Mechanisms of Long COVID—Challenges

One challenge has been the ambiguity of the host factors involved. There is a wide diversity among the affected population in age, co-morbid conditions or the severity of SARS-CoV-2 infection experienced. On one side, the affected are the survivors of moderate to severe COVID-19, most of whom, not so surprisingly, with advanced age and known risk factors such as obesity, diabetes, and hypertension [7]. It was recognized early in the pandemic that these patients require prolonged recovery times that blends into emergent Long COVID symptomology [8]. On the other side, the affected are young and previously healthy individuals with a history of mild or asymptomatic COVID-19 infection. These patients have been perplexing and recognized at large with the advent of social media and patient-reported outcomes including those from health care workers [9,10]. A recent report suggests that, similar to adults, over 10% of children can also develop symptoms of Long COVID [11]. It is reported that this ratio can reach up to 25% of children admitted during the acute COVID-19 infection [12].At present time tools for assessment of symptom severity have been limited, however, the symptoms spectrum appears to be similar across all age groups.

So far, there is no tissue diagnosis of Long COVID and the determinants of disease pathogenesis remain unclear [13,14]. Intuitively, the current trend is to search causality within the nervous system. A prospective study involving over 300 subjects with pre-existing baseline MRI data who had a recent history of mild COVID-19 has shown a pattern of changes in limbic cortical areas of gray matter along the track involving the primary olfactory and gustatory system [15]. The histopathology correlations of this observation is not known. However, post-mortem studies [16,17,18,19] -totaling over 180 COVID-19 cases with varying disease severity and time course- have depicted that SARS-CoV-2 is not a neurotropic virus. This argues against Long COVID representing a form of viral encephalitis. The presence of viral load was evident, however, in endothelial cells, astrocytes and microglia by immunohistochemistry and electron microscopy within the cerebrum, cerebellum and brain stem. Using multiplex immunostaining, Bocci et al. [20] have shown infection and disruption of pericyte homeostasis in 6 post mortem brain tissues. Perivascular inflammation was a common finding and fibrinogen leakage suggested changes in blood-brain barrier [20]. These findings were in support of an overall inclination to extrapolate microvascular dysfunction playing a fundamental role for development of Long-COVID [14,16,21]. If proven, this can account for autonomic dysfunction, postural orthostatic tachycardia syndrome (POTS), as well as cognitive impairment, fatigue and low endurance, due in part to microvasculopathy and central hypoxia [22,23,24].

In pursuit of this view, one challenge has been the paucity of biomarkers for Long COVID. Routine blood tests including complete blood count (CBC), coagulation (prothrombin time, activated partial thromboplastin time, fibrinogen), inflammation markers (C-reactive protein, interleukin-6, sCD25, ferritin) have been unrevealing [6,25]. However, recent reports allude to ongoing microvasculopathy [26,27] based on increased serum d-dimer levels in some Long COVID patients, particularly among those required hospitalization during the acute infection. In a recent study, Long COVID patients were found to have increased microclots in the peripheral blood that are large anomalous (amyloid) deposits resistant to trypsin digest and containing entrapped plasma antiplasmin as well as inflammatory mediators such as serum amyloid A (SAA) [28]. Another challenge is to identify the driving force of sustainability for vascular compromise. The current trend is to associate this with ongoing immune-dysregulation including production of a number of different autoantibodies upon exposure to cryptic antigens during SARS-CoV-2 [29]. In particular, antiphospholipid antibodies [30] are important for promoting thrombosis and endothelial damage [31]. Ongoing investigations are expected to bring more clarity on the clinical impact of autoantibodies in the long term [32].

The degree of microvacular compromise reflects the balance between the damage and repair, where damage is expected to gradually diminish and eventually resolve upon viral clearance and the down-regulation of the immune system. It has been shown that circulating endothelial cell (CEC) can be a surrogate marker to assess recovery. In the study by Chioh et al. [33], patients with cardiovascular risk factors are likely to show CEC with increased adhesion molecules and signs of activation suggestive of ongoing endotheliitis during the convalescent period post COVID-19. Furthermore, the serum content from these patients can be deprived from vascular growth factors necessary for angiogenesis. While this concept is in line with conventional wisdom, it is difficult to adapt it to Long COVID patients who experienced SARS-CoV-2 infection with minimal, if any, clinical symptoms. These minimally symptomatic patients constitute about 13–31% of all infected. Nonetheless, there is literature evidence that these patients can develop viremia, evidence of endotheliitis, and tissue tropism based on seroconversion for antiviral antibodies [34], development of pernio (i.e chilblind or covid toes) [35,36,37], and evidence of abnormal cardiac MRI for myocarditis [38], respectively. It is the expectation that these individuals should be able to repair microvascular damage swiftly, yet some of these patients suffer from severe Long COVID.

This suggest that endotheliitis may not be the only factor involved in microvasculopathy and there are additional determinants for microvascular dysfunction. In pursuit of further clarification, we suggest the central role of pericytes as discussed in the remainder of this paper. It is known that the olfactory tract provides a common route of viral entry into CNS that is often associated with a loss of smell. Based on the elegant studies using single cell expression assays the pericytes within the olfactory bub- and not the olfactory sensory neurons- have been shown to be the target cells infected with the SARS-CoV-2 virion [39]. In fact, pericytes are subject to viral tropism during SARS-CoV-2 for expressing angiotensin-converting enzyme 2 (ACE2) and transmembrane serine protease 2 (TMPRSS2) on the cell surface [20,40,41] via direct or hematogenous spread, therefore vulnerable to viral mediated damage during the viremic phase.

## 3. Our Hypothesis on Long COVID Is Based on Systems-Entropy-Information

We propose to analyze long COVID in light of general systems theory based on altered homeostasis: in this regard, the most important parameters within the open boundaries defining the state of long COVID include: absence of viral propagation, absence of rampage inflammation or autoimmune reactions, and nonlinear correlation between the severity of long COVID and reminiscent tissue damage post COVID-19. There are two key elements that are common for the transformation process: that there is altered homeostasis and reciprocity. This means change in one part reverberates throughout the system toward an“unwanted state of health”, by analogy, to increased entropy. For the process between the input (i.e., environmental insult) and output (i.e., host’s well-being) involves multiple feedback loops entangled with information processing at micro-, mezzo and macro levels. We assume that the compromise in the vascular networks is the essential element for the causality. Furthermore, unlike the classic concept of vasculopathy where changes in endothelial cells dominate, in Long COVID, we hypothesize that the changes are derived at large from altered networks of pericytes that directly influence the system’s organization and baseline stable functioning. We submit that pericytes directly impact on entropy as they are involved in distribution, analysis, and utilization of information within the system. Therefore, we suspect that the resultant chaotic and “off balance” performance of the system could be quantitatively measured using the concept of entropy.

## 4. Rationale for Expanding this View Further

Pericytes are located on the abluminal surfaces of the blood vessels tightly coupled with endothelial cells [42]. It has been well documented that the functional and structural unity between these two cell types is essential to maintain the integrity and tone of the vascular tree throughout the organism.

Genetic mutations affecting pericytes can lead to disease states as shown in preclinical models. This is particularly evident within the central nervous system [43,44,45,46,47,48,49]. It is reasonable to suggest other than genetic mutations, some acquired insults, such as viral agents or inflammatory reactions may lead to similar pathogenesis. In fact, the density of pericytes in a given unit of tissue is the highest within the brain and in the blood brain barrier [45]. This natural finding is logical as the maintenance of order (measurable by its low entropy) within the brain tissue has a strict dependence on very low tolerance to chemical or physical disturbances. Thus, any perturbance within the network of pericytes, even at sub-detection-levels, can have a perpetuating impact on brain and brain stem altering vascular tone and function that can cascade into symptoms of Long COVID as listed above.

Pericytes are found throughout tissues and constitute the population defined as mesenchymal stem cells for enfolding properties reminiscent of embryonic stem cells as they retain the ability at certain levels to promote regeneration, angiogenesis, and tropism [50,51]. These functions are necessary for life and longevity. In fact, numbers of pericytes are inversely correlated with the chronological age as shown in early studies. The yield of pericyte colonies grown ex vivo per unit bone marrow cells is 100-fold higher in infants compared to the elderly [52]. These cells have been cultured ex vivo and studied in multiple in vitro and in vivo systems in the last 3 decades as Mesenchymal Stem Cells (MSC). They have been recently regarded as Medicinal Signaling Cells, keeping the same acronym [53]. When MSCs are introduced in to a system exposed to hypoxic, chemical, physical, or inflammatory insult, they are shown to alter the landscape towards re-instituting homeostasis and reducing inflammation. They serve as an interface around the endothelial vasculature, coordinating operations and bi-directional communications in the microenvironment between the resident tissue cells and circulating blood leukocytes [53,54,55,56]. This coordination is multimodal and highly complex, owing to the plasticity and adaptability of pericyte functionality. There is an appearance of high levels of decision making. We suspect that such appearance is an emergent complex system behavior. Moreover, the mechanisms involved are evolutionary highly conserved -as evident in xenogeneic models- and based largely on paracrine factors [57]. This is further evidence of the fundamental role such emergence plays in homeostasis.

Nonetheless, there is a paucity of knowledge on network properties of pericytes in situ. This includes parameters defining its structure and function in a time continuum to predict the potential and kinetics of recovery upon an insult. Figure 1 summarizes our view that Long COVID is likely to be a multifaceted and cascading process with a common element of cumulative changes in the pericyte network and microvascular dysfunction.

## 5. Triple A Concept-New Disease Model-Illustrating Human Biology

Within this scheme, we like to introduce the concept of “pericyte disarray” to discuss how an altered state of pericytes may be the primary culprit to Long COVID. We argue that this may be a new disease category that hitherto has not been clearly defined in medicine. In the process, we like to propose the assumption that the pericytes have the unique ability to *Analyze*, *Adapt* and *Advance* (“Triple A”) the microenvironment, acting as system integrators at the interface of the resident tissue and the circulating blood content within the vascular system. This eventually results with information flow at cellular, microenviromental, and organism levels constituting the transformation process between a harmful insult to a host and recovery-survival-regeneration.

The preponderance of preliminary evidence points to this promising hypotheses if the Triple A is addressed as a distributed information processing mechanism that governs and regulates systemic functions. We propose to build quantitative models to describe the level of regulation versus the state of off-balance behavior by measuring the entropy of the systems. Being distributed would attempt to explain how Long COVID seems recalcitrant to pinpointing its source cause. The study and any discovery would necessitate a systems view which is best tested by constructing complex dynamical models and studying the resultant (possible agent-based) simulations for their likeness to observed phenomena. As a novel step, we posit that not only the regulatory mechanisms are distributed, as seen in many biological functionalities, but also the embedded “decision-making” processes are distributed, hence contributing to the elusive nature of Long COVID.

Therefore future work will focus on identifying this hitherto unknown network of information processing and distributed control and collaboration among pericytes. In short, we hope to discover a network of activity among system wide pericytes as the bio-cosmic web and suspect that any disarray in the information processing of this distributed system is responsible for the variety of elements common in long COVID, and perhaps in similar maladies such as the Gulf War Syndrome and the Havana Syndrome. Since the nature of proposed research deals with a distributed information processing system, the work will necessarily follow the tools, techniques, methodology, and approaches in systems theory with a focus on complex dynamical systems. There exist many tools at our disposal from general systems theory [58]. In fact, most of the theoretical work has its origins in medical works, best exemplified by Tektology [59,60]. Recent advances in computer simulation as well as the recent interest in complexity theory also provide us with tools and road-maps towards modeling and examining, measuring, and evaluating systems behavior.

Figure 2 exemplifies our inclination to study the phenomenon as a system. Building on mid-20th Century work on systems, with models involving stocks and flows [61,62], we are mindful of the so-called endogenous viewpoint.

Rather than a singular central cause of Long COVID, we take the view that the structure of a system, having evolved to function while in balance (homeostasis), when disturbed, and in extreme cases, when in disarray, may give rise to the observed symptoms. We posit that being systemic, the pathogenesis remains elusive, and is best treated as an emergent property. Its understanding would thus benefit from a systems study. In short, the causes may be systemic, emergent, and from within, rather than singular and without.

Systems dynamics simulations are thus seen as a tool to further investigate the phenomenon. As Richardson [63] puts it,


*“System Dynamics is the use of informal maps and formal models with computer simulation to uncover and understand endogenous sources of system behavior.”*


We envision a gradual construction of a systems model, inserting the available relationships as we test to see if the models display similar emergent behavior. The work is also useful in identifying potential elements and interactions that may play a role in the emergent ailment which have hitherto not fully explored.

## 6. Conclusions

This hypothesis paper presents a generic viewpoint that Long COVID as well as other diseases may be the cause of a systemic disarray of pericytes. Evidence is presented that supports this hypothesis. Moreover, the system, albeit distributed in nature, is suspected to be capable to *Analyse*, *Adapt*, and *Advance* (the Triple-A). The system collects and processes information, and then generates a customized response. Thus, we propose that the best way to test the hypothesis is to first develop a systems model and express the order or disarray quantitatively using concepts of entropy, agent-based modeling, and nested hierarchies of systems that are capable of generating emergent behavior. By its nature, this paper only offers a general hypothesis that presents a viewpoint, a framework, and suggests tools. Only data in existing literature are reffed to. Specific models are currently being developed. These models are expected to give rise to testable cases, where data will be collected and analyzed for its agreement or disagreement with the hypothesis. These models and further experiments are being pursued and their outcomes will be reported subsequently.

## Figures and Tables

**Figure 1 jcm-11-00572-f001:**
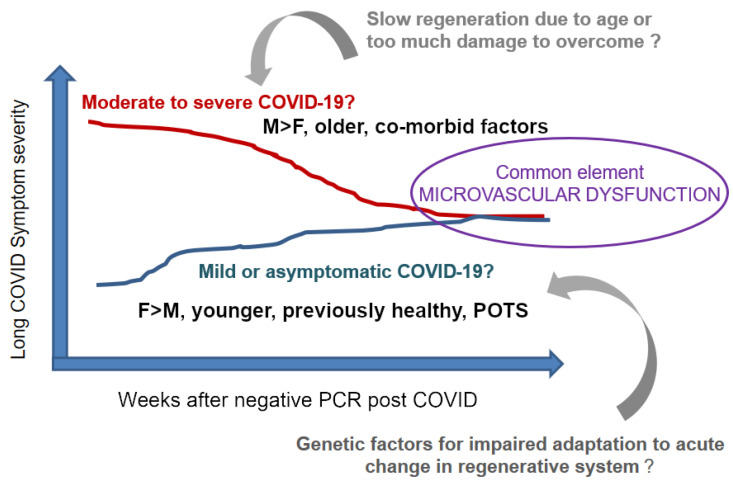
We envision Long COVID involves microvascular dysfunction due in part to suboptimal recovery of pericytes from viral mediated damage. There are likely to be several root-causes: one, from escalated vascular damage during acute COVID-19 on top of existing microvascular changes particularly in the presence of age associated limitations of regeneration (red line). Another speculation is genetic predisposition to breakdown of pericyte networks upon external insult (blue line) that may include mutations affecting pericyte structure and functions. Males (M) over females (F), the older population, or in the presence of co-morbid factors are more likely to be in the former group. The latter group is more likely to contain females, the younger, the previously healthy, or with postural orthostatic tachycardia syndrome (POTS).

**Figure 2 jcm-11-00572-f002:**
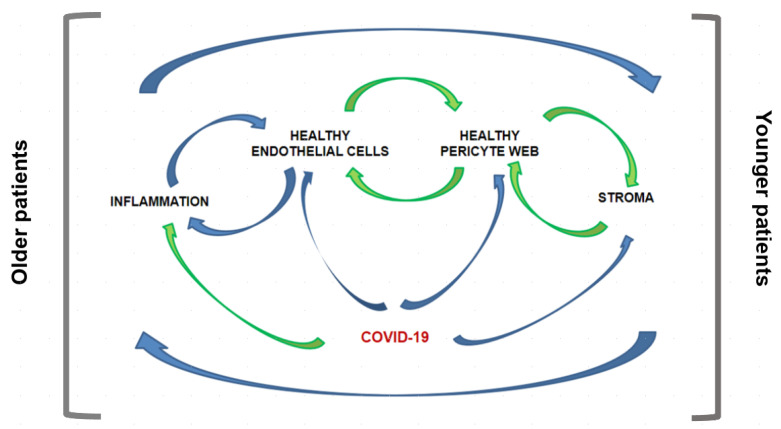
The casual loop proposition of hypothesis presented. Blue line for negative feedback and green line for positive feedback. Root-causes of altered pericyte networks may vary under the influence of different sets of genetic or epigenetic factors, and along the arrow of time.

## Data Availability

Not applicable.

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
