# Peer review of "Is Long COVID a State of Systemic Pericyte Disarray?"

_jcm, 2022, doi:10.3390/jcm11030572_

Round 1
Reviewer 1 Report
The authors hypothesize based an available literature that Long Covid is -at least partially-cased by a disarray of pericytes. The support this view with relevant citations and suggest, to explore this further using a model to integrate system theory and complex adaptive systems.
This is an interesting hypothesis that very well warrants further exploration. How this in detail can be done using the suggested approach does not become clear from the article. The commentary expresses rather an intention to study this causality, but does not provide any data. The quality of literature review and reasoning is, however, adequate.
The usefulness of such an article within the scope of the journal lies with the editors.
Minor comment: reference 19 _ citation format in bibliography needs to be changed
Author Response
We thank the reviewer. It is entirely true that the current paper does not have any new data other than reviewing and acknowledging data from existing literature. The current paper attempts to make sense of the available information, observations, and data to suggest a hypothesis and propose a direction of research. Data and specific models to ascertain the validity of the hypothesis will be forthcoming as ongoing work matures.
We have included comments regarding this point to clarify the nature of the paper. We once again thank the reviewer, whose comments have lead to further clarification and improved the readability of the paper.
We thank the reviewer for pointing out citation 19. We used the MDPI Latex template in its default configuration. Apparently the template seems to use only initials if there are more than 10 authors. The guide (mdpi_references_guide_v5.pdf) indicates that all authors may be listed, even if there are more than 10. We will consult with the editor and correct as needed.
Reviewer 2 Report
The paper presents a generic viewpoint that Long COVID as well as other diseases may be the cause of a systemic disarray of pericytes. The work deals with an extremely interesting, important and contemporary topic, that of the COVID outbreak. The authors have based their work on an extensive literature, most of which is from year 2021. Findings are very clearly presented and supported by data.
I would welcome more elaboration on the future work needed and some extensions of this research.
It will contribute to the efforts regarding research on COVID.
Author Response
We thank the reviewer. We appreciate the comment “I would welcome more elaboration on the future work needed and some extensions of this research.” which is precisely our intention with ongoing work. We would welcome all forms of future collaboration with the readership to tackle this very important subject.